# Grape Seed Proanthocyanidins Improve the Quality of Fresh and Cryopreserved Semen in Bulls

**DOI:** 10.3390/ani13172781

**Published:** 2023-08-31

**Authors:** Meng Wang, Silin Wu, Benshun Yang, Miaomiao Ye, Jianbing Tan, Linsen Zan, Wucai Yang

**Affiliations:** College of Animal Science and Technology, Northwest A&F University, Yangling 712100, China; wangmeng1001@nwafu.edu.cn (M.W.); wsliiiing@163.com (S.W.); ybs2820807719@163.com (B.Y.); 17331349676@163.com (M.Y.); 13187279307@163.com (J.T.); zanlinsen@163.com (L.Z.)

**Keywords:** GSPs, bull semen, cryopreservation, fresh semen quality, antioxidant activity

## Abstract

**Simple Summary:**

As the basis of artificial insemination technology, semen cryopreservation technology is a very valuable technology in the cattle industry and directly affects the success rate of fertilization. The effectiveness of semen cryopreservation is closely associated with quality alterations of fresh semen and semen diluent during semen cryopreservation. Therefore, in this study, in order to improve the effect of semen cryopreservation and fresh semen quality of breeding bulls, enough antioxidant-efficient grape seed proanthocyanidins were added to semen diluent and breeder bull diets. It was found that grape seed proanthocyanidins as a semen diluent extender could improve frozen–thawed sperm morphology and quality (including frozen–thawed sperm motility, membrane integrity, acrosome integrity, and mitochondrial activity). As a dietary additive, it could improve sperm motility, reduce sperm deformity rate, and improve the quality of fresh semen. This study revealed that grape seed proanthocyanidins could improve effect of semen cryopreservation and the reproductive performance of breeding bulls, which was significantly related with the antioxidant properties of grape seed proanthocyanidins. This research promotes the preservation and breeding of bovine semen and provides a basis for the fundamental science of mammalian gametes and reproductive biotechnology.

**Abstract:**

Oxidative stress leads to a decrease in semen quality during semen cryopreservation and fresh semen production. Grape seed proanthocyanidins (GSPs) are endowed with well-recognized antioxidant, anti-inflammatory, anti-cancer, and anti-aging activities. Therefore, the objective of this experiment was to explore the effects of GSPs on the quality of fresh and cryopreserved semen to provide a basis for GSPs as a new dietary additive and semen diluent additive for males’ reproduction. Fresh semen from three healthy bulls aged 3 to 5 years old were gathered and mixed with semen diluents dissolved with 0 µg/mL, 30 µg/mL, 40 µg/mL, 50 µg/mL, and 60 µg/mL GSPs respectively. The motility, physiological structures (acrosome integrity, membrane integrity, mitochondrial activity), and antioxidant capacity of frozen–thawed sperm were measured after storage in liquid nitrogen for 7 days (d). Bulls were fed with 20 mg/kg body weight (BW) GSPs in their diet for 60 days; the weight of the bull is about 600 kg. Then, the reproductive performance and antioxidant indexes of bulls were measured before and after feeding. The results demonstrated that GSPs supplementation significantly increased sperm motility, physiological structures, GSH-Px, and CAT enzyme activities and significantly decreased MDA content in sperm during semen cryopreservation. The optimal concentration of GSPs was 40 µg/mL (*p* < 0.05). After 20 mg/kg (body weight) GSP supplementation, sperm motility was significantly heightened (*p* < 0.05), the sperm deformity rate was significantly reduced (*p* < 0.05), and antioxidant enzyme activities (such as SOD, CAT, and GSH-Px) were significantly enhanced (*p* < 0.05), and the production of MDA was significantly suppressed (*p* < 0.05) in serum compared with that before feeding. In conclusion, these results reveal that a certain concentration of GSPs has a good protective effect on sperm damage caused by semen cryopreservation and the reproductive performance reduction caused by stress in bulls, which may be attributed to the antioxidant function of GSPs. In summary, GSPs are a useful cryoprotective adjuvant and dietary additive for bull sperm quality.

## 1. Introduction

Reactive oxygen species (ROS) is identified as an imbalance standing between reactive oxygen species (ROS) production and the ability to readily detoxify these reactive intermediates or to easily repair the resulting damage [1]. ROS are generated and required during physiological processes regarding the structural and functional maturation of spermatozoa, whereas ROS trafficking between mitochondria could constitute a positive-feedback mechanism resulting in an elevated production of ROS that could be propagated throughout the cell and may cause perceptible mitochondrial and cellular injury [2]. Meanwhile, pathologically strengthened ROS levels have been repeatedly associated with male reproductive dysfunction [3,4]. Spermatozoa are particularly sensitive to ROS because their plasma membranes have high polyunsaturated fatty acids, which are the main targets of oxidation, while their cytoplasm is mainly restricted to the midpiece with very few antioxidant mechanisms to provide adequate protection against oxidative damage [2,5]. Excessive ROS are produced in the sperm cells during temperature stress in the process of cryopreservation of semen and may lead to lipid peroxidation (LPO), DNA fragmentation, the alteration of enzymatic pathways and cellular communication, which in proper order are strongly correlated with motility loss, damage to sperm membrane integrity, poor fertilization rates, or impaired embryogenesis [6,7]. During the breeding process of male animals, many factors such as high ambient temperature, restricted activity space, and high grain-to-grain ratio can cause oxidative stress in the reproductive organs of male animals, which in turn leads to the decline of semen quality and reproductive performance [8]. Therefore, it is necessary to reduce the oxidative damages induced by sperm freeze–thawing and the sperm formation of the body.

In recent years, numerous researchers have displayed that the in vitro disposal of hydrophilic or lipophilic antioxidants in human or veterinarian andrology has a significant influence on critical semen variables including sperm motility, sperm membrane and sperm DNA integrity, and reducing the oxidative damage of sperm cryopreservation [9,10]. Moreover, in vivo antioxidants may protect spermatozoa from ROS produced by leukocytes, block premature sperm maturation, and provide overall stimulation to the male gamete [11]. There were studies that a higher nutrient intake of vitamins C, E, and b-carotene was associated with a higher sperm count number and motility [12].

Grape seed proanthocyanidins (GSPs) are a class of polyphenol compounds with active functions which can be found in grape branches, leaves, peel, pulp, or grape seeds [13,14]. GSPs are a new type of high-efficiency antioxidant with the characteristics of safety, novelty, and bioavailability; they have been displayed to quench singlet oxygen 50 times as efficiently as vitamin E and 20 times compared with vitamin C [15]. So far, GSPs are one of the most powerful free radical scavengers, which have very strong activity in vivo, and can effectively eliminate superoxide anion free radicals and hydroxyl radicals [16]. In addition, GSPs also have other significant biological activities, including anti-tumor and anti-cancer activities [17] as well as the prevention and treatment of heart diseases, type 2 diabetes [18], and so on. Therefore, GSPs are widely applied in functional foods, health supplements, and clinical nutrition [19]. A growing number of researchers are highlighting the conducive roles of GSPs supplementation in the antioxidant ability of different tissues. Studies have shown that adding an appropriate concentration of GSPs can significantly improve the semen cryopreservation effect of pigs [20,21]. Moreover, a study has shown that GSPs could alleviate oxidative stress in vivo [22], and dietary GSPs could prevent ultraviolet-radiation-induced non-melanoma skin cancer through the enhanced repair of damaged DNA-dependent activation of immune sensitivity [23]. In addition, in vitro and vivo malformation experiments in mice proved that GSPs do not have potential genotoxicity but have certain anti-mutagenic properties, which can effectively reduce the mitochondrial mutation rate [24].

However, it has not been clarified whether GSPs affect the quality of fresh and cryopreserved semen in bull. This study preliminarily explored the role of GSPs in the semen cryopreservation and fresh semen production process.

## 2. Materials and Methods

All protocols of this experiment in our study were completely approved by the Laboratory Animal Management Committee of Northwest A&F University (Yangling, China).

### 2.1. Extender Preparation

The extender was prepared using a solution consisting of 1.354 g sodium citrate (RHAWN, R014169-500g, Shanghai, China), 2.4224 g TRIS (Solarbio, T8060-500g, Beijing, China), 1.1 g glucose (BGI, 1.18392.010, Shenzhen, China), 5.33 mL glycerol (Solarbio, G8190-100mL, Beijing, China), 20 mL egg yolk, 0.587 mL penicillin-streptomycin solution (HyClone, UT, USA), and 68.82 mL double distilled water, and the mixture was thoroughly through shaking [25,26] and then stored at 4 °C. Next, 1 mg of GSPs (Sigma, 29106-49-8, Darmstadt, Germany) was dissolved in 1 mL basal dilution to form a concentration of 1 mg/mL GSPs working solution, and then 0 mL, 0.15 mL, 0.2 mL, 0.25 mL, 0.3 mL of GSPs working solution were added to 5 mL of basal dilution, respectively. The concentrations of the GSPs are 0 µg/mL, 30 µg/mL, 40 µg/mL, 50 µg/mL, 60 µg/mL, respectively.

### 2.2. Semen Collection and Semen Quality Analysis

Fresh semen was collected from three healthy bulls aged 3–5 years old using an artificial vagina. Sperm was collected twice a week, and ejaculation was performed once. The collected semen was transported to the research laboratory at 37 °C, and the semen quality was immediately analyzed using a computer-associated sperm analysis (CASA) system (Israélien). Various parameters related to sperm quality are as follows: sperm motility (%), progressive motility (%), motile sperm concentration (million/mL), progressive motility sperm concentration (million/mL), and average velocity of progressive sperm (microns/s).

### 2.3. Sperm Freezing and Sperm Thawing

Semen with sperm motility ≥65%, sperm density ≥6 × 10^8^ mL, and sperm malformation rate ≤15% were diluted via a one-step method according to a certain proportion. The preheated bovine semen diluent was slowly poured into fresh semen (fresh semen was always kept at 37 °C before dilution), and the diluted semen was divided using a fine tube filling and sealing machine. After 3 h equilibrium in the refrigerator at 4 °C, the temperature of semen was slowly lowered to −145 °C in a 4 °C freezer with a programmed automatic freezer, and then the 0.25 mL tubes of semen were quickly put into liquid nitrogen. After 7 days, the 0.25 mL tubes were quickly taken out from the liquid nitrogen and thawed in a 37 °C water bath for 30 s for follow-up testing.

### 2.4. Detection of Sperm Plasma Membrane Integrity

The sperm membrane integrity was detected using the Hypo-osmotic Swelling Test (HOST) method. The hypotonic solution was prepared according to the proportion, adding 100 µL of thawed semen to 1 mL of isothermal hypotonic solution preheated in advance, mixing well and incubating for 30 min, drawing 10 µL to make a glass slide, and observing under a 400× microscope, with the sperm plasma membrane bent at an angle showing intact. According to “the rate of sperm bending tail (%) = the number of sperm in the curved tail/the total number of sperm counted × 100”, the rate of bending tail is counted, and at least 200 sperms are counted each time, repeated more than 3 times.

### 2.5. Detection of Sperm Acrosome Integrity

The sperm acrosome integrity was detected by means of fluorescent labeling via peanut agglutinin (FITC-PNA) staining (sigma, Darmstadt, Germany). An amount of 30 µL of thawed semen was aspirated, spread evenly on a clean glass slide, air dried naturally, fixed with 4% tissue cell fixative for 20 min, stained with 10 µL FITC-PNA dye, and stored in a dark and humidified box at 37 °C. It was rinsed with PBS 3 times, and pictures were taken under a fluorescence microscope at 400× after incubation for 30 min. The acrosome is complete if the head shows green fluorescence, that is, “acrosome integrity (%) = the number of sperm with green fluorescence in the head/total number of sperm counted × 100”, each count was performed at least 200 times and repeated more than 3 times.

### 2.6. Detection of Sperm Mitochondrial Activity

Rhodamine (Rh123) fluorescent dye staining (Sigma, Saint Louis, MO, USA) was used to distinguish whether sperm mitochondria were active or not, because Rh123 can specifically bind to well-functioning mitochondria, making sperm tails emit green light under a fluorescence microscope, and the head does not glow but the tail has green fluorescence. First, 200 µL of thawed semen was added to 500 µL of pre-warmed isothermal diluent; 3 µL of Rh123 was added to pre-warmed 200 µL of PBS buffer and incubated at 37 °C for 10 min in the dark to form a staining solution. Next, 100 µL of diluted semen was incubated in water at 37 °C for 30 min, and a 10 µL sample was drawn to make a glass slide, which was observed under a fluorescence microscope at 400×; mitochondrial activity was calculated according to the following: “mitochondrial activity (%) = green fluorescence in the tail but no light in the head sperm count/count total sperm count × 100”. At least 200 sperm were counted each time and repeated more than 3 times.

### 2.7. Dietary Feeding Test

Bulls aged 3–5 years old were fed with a basal diet supplemented with GSPs (20 mg/kg body weight). The weight of the three bulls was about 600 kg, and the basal diet of the bulls remained unchanged before and after feeding GSPs (Senfu Biotechnology Co., Ltd., Xian, China), with drinking freely. The pre-experiment lasted for 15 days, and the formal experiment lasted for 45 days. Fresh semen and blood were collected before and after feeding GSPs. The quality of fresh semen was immediately analyzed, and the serum was separated and stored at −80 °C until examination.

### 2.8. Antioxidant Analysis in Serum and Seminal Plasma

Semen samples of 100 µL were incubated at 37 °C for 30 s and centrifuged at 1600 r/min for 5 min, and precipitate was collected after centrifugation. The precipitate, supplemented with 200 µL solution, was treated with 200 µL of TritonX-100 (1%) for 20 min, and centrifugation was performed at 4000 r/min for 20 min; the supernatant used for the extraction of enzymes in semen was collected. Then, the malondialdehyde (MDA) content and activities of catalase (CAT) and glutathione peroxidase (GSH-Px) in semen were measured using an enzyme activity assay kit (Nanjing Jiancheng Biological Company, Nanjing, China). Likewise, the activities of CAT, GSH-Px, and superoxide dismutase SOD enzyme as well as the content of MDA were measured.

### 2.9. Statistical Analysis

Statistical analysis was carried out using the GraphPad Prism program and IBM SPSS statistics. Tukey’s test was applied to a post hoc test, and the one-way analysis of variance procedure was used as the comparison of the mean value of the sperm kinematic parameters and enzymatic activity. All results were presented as mean ± standard deviation, and *p* < 0.05 was considered significant.

## 3. Results

### 3.1. Effect of Supplemented GSPs Extender on Sperm Motility and Progressive Velocity in Semen Collected with Artificial Vagina

The effect of GSPs in the freezing extender on the motion characteristics of cryopreserved sperm are shown in Table 1. The motility and progressive motility of frozen–thawed sperm in bull were significantly improved (*p* < 0.05) with the presence of GSPs in the extender except at the concentration of GSPs was 60 μg/mL, as compared with the control. The motility and progressive motility of frozen–thawed sperm supplemented with 40 µg/mL and 50 µg/mL were significantly higher than that of other concentrations (*p* < 0.05), and had better performance in terms of motile sperm concentration, progressive motility sperm concentration, and the velocity of progressive motility sperm, though the differentiation is not significant (*p* > 0.05). These results indicated that 40 µg/mL and 50 µg/mL GSPs supplementation significantly improved (*p* < 0.05) sperm motility and movement characteristics.

### 3.2. GSPs Protect the Membrane Integrity of Frozen–Thawed Sperm

Hypotonic solutions showed that the sperm plasma membrane integrity, swelling of sperm, and convolution of the tail of the sperm represented plasma membrane integrity but were unchanged in the incompleteness of plasma membrane integrity; the sperm plasma membrane integrity of the picture is shown in Figure 1A. The results of the tests of membrane integrity on frozen–thawed bovine sperm are shown in Figure 1B. Compared with the control group, the membrane integrity of frozen–thawed sperm supplemented with 30 µg/mL, 40 µg/mL, and 50 µg/mL were significantly improved (*p* < 0.05), and the optimum GSPs concentration in bovine semen extender was 40 µg/mL.

### 3.3. GSPs Protect Acrosome integrity of Frozen–Thawed Sperm

Heads with green fluorescence presented acrosome integrity, which is shown in Figure 2A. The results of the tests of acrosome integrity on frozen–thawed bovine sperm are shown in Figure 2B. Compared with the control group, the acrosome integrity of frozen–thawed sperm supplemented with 30 µg/mL, 40 µg/mL, and 50 µg/mL were significantly improved (*p* < 0.05), and the optimum GSPs concentration in bovine semen extender was 40 µg/mL.

### 3.4. GSPs Protect Mitochondrial Activity of Frozen–Thawed Sperm

A tail with green fluorescence but a head without green fluorescence shows mitochondrial activity, which is shown in Figure 3A. The results of the tests of mitochondrial activity on frozen–thawed bovine sperm are shown in Figure 3B. Compared with the control group, the mitochondrial activity of frozen–thawed sperm supplemented with 30 µg/mL, 40 µg/mL, and 50 µg/mL was significantly improved (*p* < 0.05), and the optimum GSPs concentration in bovine semen extender was 40 µg/mL.

### 3.5. GSPs Increase Sperm Oxidation Resistance of Frozen–Thawed Sperm

The sperm oxidation resistance of frozen–thawed sperm treated with different concentrations of GSPs is summarized in Table 2. The freezing media supplemented with 30 µg/mL, 40 µg/mL, and 50 µg/mL GSPs had a significantly beneficial effect on the activities of antioxidant enzymes of catalase (CAT) and glutathione peroxidase (GSH-Px) (*p* < 0.05) and significantly decreased malondialdehyde (MDA) content (*p* < 0.05), and the optimum GSPs concentration in bovine semen extender was 40 µg/mL. However, when the concentration of GSPs was increased to 60 µg/mL, there was partial inhibition effect in the antioxidant properties of frozen–thawed sperm.

### 3.6. GSPs Elevate Fresh Semen Quality after Dietary Feeding

Good fresh semen quality is the foundation of semen cryopreservation. The evaluation of fresh semen quality includes the detection of sperm motility and the sperm deformity rate, while male diets are one of the main factors affecting the quality of fresh semen. To some extent, the application of dietary additives could decrease the reduction in reproductive performance caused by stress in breeding male bulls and protect sperm from damaging. Therefore, we detected the changes in the semen quality of breeding bulls before and after feeding GSPs in Table 3. The results showed that in 20 mg/kg BW GSPs-fed bulls, the motility and progressive motility significantly increased (*p* < 0.05) after feeding GSPs, and the sperm deformity rate had significantly reduced (*p* < 0.05). We concluded that a dietary supplement of GSPs could improve the semen quality of breeder bulls, thus improving the reproductive performance of breeder bulls.

## 4. Discussion

A certain degree of irreversible sperm damage is caused due to oxidative stress in the process of semen generation and freezing; this damage seriously affects the effect of semen cryopreservation and restricts the development of animal husbandry. Because of their superior antioxidant capacity, GSPs push forward an immense influence on sperm oxidative stress damage caused by semen freezing and fresh semen production.

GSPs play an important protective role in various oxidative-stress-induced injuries and diseases of the body. A case in point is that GSPs could protect the epidermal keratinocytes of humans from extensive damage to ultraviolet irradiation [27]. Oxidative-stress-induced spermatozoa damage is also one of the most important causes of sperm cell damage during the semen cryopreservation and semen thawing process [28]. Frozen–thawed sperm is more easily peroxidated than fresh sperm; intracellular antioxidant capacity fails to provide protection against oxidative damage and the potential toxic effects of free radicals following freeze–thawing [29]. These toxic effects include reducing sperm motility through disrupting the physiology integrity and physicochemical properties of sperm, such as damage to membrane structures and a decrease in mitochondrial activity [2]. In this regard, the fortification of the extender with an exogenous antioxidant is necessary to shield bull spermatozoa from the stressful effects of cryopreservation [2,10,30]. Some studies have confirmed that antioxidants (such as vitamin E, glutathione, N-acetylcysteine, catalase, and ferulic acid) are effective at reducing ROS levels and preventing a decline in sperm motility during semen cryopreservation [31,32]. There was also a study about melatonin as an antioxidant that could protect goat semen against oxidative damage during cryopreservation through improving antioxidant capacity [33]. In this study, different concentrations of GSPs were added to the semen diluent; these frozen–thawed sperm indicators were detected, including the semen motility, viability, mitochondrial activity, plasma membrane integrity, acrosome integrity, and antioxidant potential. It was found that GSPs had a protective effect on semen motility, viability, mitochondrial activity, plasma membrane integrity, acrosome integrity, and antioxidant status. These findings are consistent with a previous study [21], further confirming the antioxidant effect of GSPs on the semen cryopreservation process.

Mammalian spermatogenesis is a complex process, which proceeds through a well-defined order including the mitotic expansions of spermatogonia, the meiotic reduction divisions of spermatocytes, and spermiogenesis [34,35]. The total duration cycle of bull spermatogenesis is 61 days [36]. Compared to sperm, spermatogonia are more highly sensitive to oxidative stress among spermiogenesis. Therefore, oxidative stress should be reduced as much as possible during semen production to improve semen quality. Breeding management, such as the use of feed additives, can affect sperm quality in males during semen production [37]. We further explored the protective effects of GSPs through adding GSPs to the diet of bulls. In this study, we investigated the effects of GSPs on bull antioxidant status, comparing the semen quality, SOD activity, CAT activity, GSH-Px activity, and MDA before and after feeding. This result uncovered that the supplementation of GSPs effectively relieved the sperm deformity rate but promoted sperm motility and the antioxidant properties of the blood. Similar results were verified in the experiments of Wang et al. Wang et al. found that the dietary supplementation of GSPs could improve the antioxidant status and hormone levels in serum, thereby improving reproductive performance in the colostrum of multiparous sows [38]. The study by Jaime M et al. indicated that the higher the intake of antioxidant nutrients like vitamins C, E, and 3-carotene, the higher the semen quality, including sperm count number and motility in patients attending a fertility clinic [12]. Previous studies also showed that GSPs could mitigate cisplatin and arsenic-induced oxidative stress in research on rodents [22,39]. Especially given the ban on antibiotics as feed additives, phytogenic feed plays an important role in feed additives due to its antioxidant and anti-inflammatory characteristics; this includes ginger, green tea, echinacea extract, purslane extract, honey suzuki, moringa and rosemary, and other phenolic compounds [40,41]. On this basis, this study provided a basis for improving the reproductive performance of breeding animals with phytogenic feed additives [37].

Excess reactive oxygen species (ROS) molecules instigate severe damages induced via oxidative stress to the quality parameters of sperm bull during semen cryopreservation [4,31,42]. GSPs are one of the proanthocyanidins currently known as the most efficient free radical scavengers in plant extracts [27]. GSPs can directly neutralize ROS and inhibit the formation of the oxidation product malondialdehyde (MDA) to reduce oxidative stress damage [1,27,43]. In this study, the sperm content of MDA was significantly reduced in the 30, 40, 50, 60 µg/mL GSPs compared with the control in frozen–thawed semen. Several studies have shown that GSPs inhibit oxidative stress damage through the activation of mitogen-activated protein kinases (MAPKs) and NF-κB signaling [35], while mammalian target of rapamycin (mTOR) is necessary for both the proliferation and differentiation of progenitor spermatogonia during spermatogenesis [44,45]. A previous study indicated that GSPs can improve chemotherapy drug cisplatin (CIS)-induced testicular cell apoptosis via activating the phosphatidylinositol 3-kinase (PI3K)/V-akt murine thymoma viral oncogene homolog (Akt)/mTOR signaling pathway [46]. Although the in vitro experiments of this study have different triggers for oxidative damage compared to previous studies, we suspect that the antioxidant effect of GSPs may be through the activation of the MAPK, NF-κB, and PI3K/Akt/mTOR signaling pathways. In addition, studies have also shown that GSPs can also reduce lipid peroxidation and increase testis antioxidant activity through increasing mitochondrial activity [47], and the enhanced effect of GSPs on mitochondrial activity was also confirmed in our study. Furthermore, a study of GSPs supplementing the diet of pigs has shown that GSPs feeding caused ecological changes in the gut microbiome and significantly increased the number of intestinal core flora, such as Lachnospiraceae, Clostridales, Lactobacillus, and Ruminococcacceae [48]; this suggests that GSPs may also improve the physical health status of breeding bulls through increasing the number of intestinal core bacteria, further improving semen quality in this study.

While GSPs are a good dietary additive that can improve the reproductive performance of breeding animals, GSPs contain anthocyanins, which will affect the appetite of breeding animals, so the amount of GSPs should be strictly controlled [16].

Moreover, the current study was limited by the lack of bull of sperm with cryopreservation after feeding GSPs and an in vivo fertility assessment to confirm the protective effect of the supplementation of GSPs on the cryopreservation efficiency of bull semen. Frozen semen is for artificial insemination, and the ultimate goal of improving the quality of semen freezing and fresh semen is to improve the fertility rate of female animals, so it is necessary to track the fertility rate of frozen semen containing GSPs additives after artificial insemination to more accurately verify GSP-induced improvement on reproductive performance. The testis is the site of spermatogenesis, so it is essential to study the weight and volume shifts of the testicular epididymis in breeding bulls before and after feeding them GSPs. We sincerely hope that the limitations of the present study will inspire more research to determine the potential effect of GSPs on the reproductive performance of breeding bulls.

## 5. Conclusions

In conclusion, this study uncovered GSPs as an exogenous additive to bull semen diluent that could protect spermatozoa from cryopreservation injuries, as evidenced by frozen–thawed sperm viability, motility, antioxidant enzyme activity, membrane integrity, mitochondrial activity, and acrosome integrity, with 40 µg/mL being the most effective. GSPs were added to the diet of bulls to test the semen quality and blood antioxidant properties of bulls before and after feeding this antioxidant, and this study confirms GSPs could improve semen quality through enhancing the antioxidant capacity of bulls.

## Figures and Tables

**Figure 1 animals-13-02781-f001:**
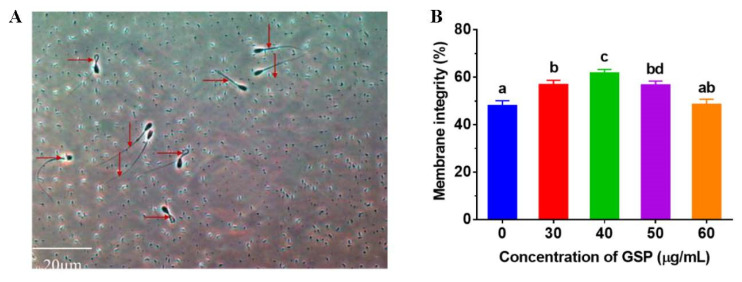
(**A**) Images of sperm membrane integrity. “→” indicates sperm with integral membrane; “↓” indicates sperm with damaged membrane. (**B**) Effect of different concentrations of GSPs on sperm membrane integrity of frozen–thawed sperm in bull. Notes: Values with same lowercase superscript in the same column mean no significant difference (*p* > 0.05), but values with different lowercase superscript mean significant difference (*p* < 0.05).

**Figure 2 animals-13-02781-f002:**
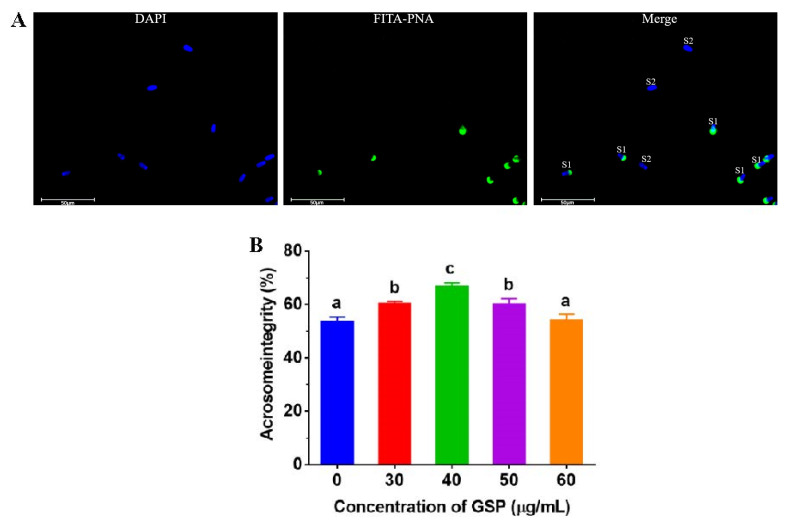
(**A**) Images of sperm acrosome staining. “S1” indicates sperm with integral acrosome; “S2” indicates sperm with damaged acrosome. (**B**) Effect of different concentrations of GSPs on sperm acrosome integrity. Notes: Values with same lowercase superscript in the same column mean no significant difference (*p* > 0.05), but values with different lowercase superscript mean significant difference (*p* < 0.05).

**Figure 3 animals-13-02781-f003:**
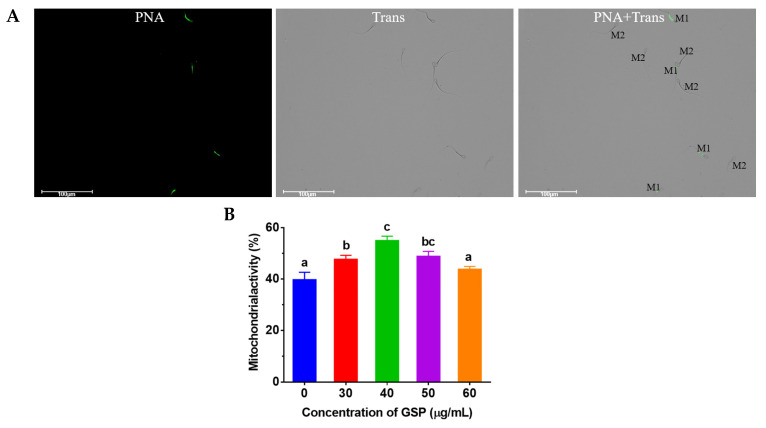
(**A**) Images of sperm mitochondrial staining. “M1” indicates sperm with mitochondrial activity; “M2” indicates sperm without mitochondrial activity. (**B**) Effect of different concentrations of GSPs on sperm mitochondrial activity Notes: Values with same lowercase superscript in the same column mean no significant difference (*p* > 0.05), but values with different lowercase superscript mean significant difference (*p* < 0.05).

**Table 1 animals-13-02781-t001:** Effect of different concentrations of GSPs on sperm parameters of frozen–thawed semen.

GSPs. (µg/mL)	Motility (%)	PR.MOT (%)	MSC (M/mL)	PSCM (M/mL)	Velocity (mic/s)
0	45.80 ± 0.71 ^c^	35.55 ± 0.49 ^b^	79.40 ± 6.22 ^a^	61.60 ± 4.67 ^ab^	34.50 ± 2.12 ^ab^
30	51.23 ± 1.44 ^b^	39.77 ± 1.01 ^c^	90.60 ± 11.38 ^ab^	70.30 ± 8.59 ^ab^	33.00 ± 3.46 ^b^
40	57.80 ± 1.45 ^a^	45.23 ± 1.25	91.07 ± 18.81 ^ab^	71.27 ± 14.76 ^a^	37.67 ± 1.15 ^a^
50	55.80 ± 1.22 ^a^	43.5 ± 0.79 ^a^	91.80 ± 3.30 ^b^	71.60 ± 2.77 ^a^	35.00 ± 0.00 ^ab^
60	44.43 ± 1.92 ^c^	39.96 ± 4.54 ^b^	71.73 ± 4.31 ^b^	55.33 ± 3.21 ^b^	31.67 ± 1.53 ^b^

**Notes:** Values with same lowercase superscript in the same column mean no significant difference (*p* > 0.05), but the values with different lowercase superscript mean significant difference (*p* < 0.05). PR.MOT = progressive motility; MSC = motile sperm concentration; M = million; PSCM = progressive motility sperm concentration. The same below.

**Table 2 animals-13-02781-t002:** Effects of different concentrations of GSPs on sperm oxidation resistance of frozen–thawed semen in bull.

GSPs. (µg/mL)	CAT (U/mg Protein)	GSH-Px (U/mg Protein)	MDA (nmol/mg Protein)
0	1.70 ± 0.07 ^d^	30.92 ± 0.04 ^d^	3.77 ± 0.04 ^a^
30	3.29 ± 0.09 ^b^	47.03 ± 3.32 ^c^	2.05 ± 0.18 ^b^
40	4.81 ± 0.09 ^a^	69.13 ± 2.48 ^a^	1.24 ± 0.02 ^c^
50	2.72 ± 0.10 ^c^	57.18 ± 2.75 ^b^	1.39 ± 0.04 ^d^
60	1.56 ± 0.07 ^d^	23.66 ± 1.08 ^e^	1.67 ± 0.07 ^e^

Notes: Values with same lowercase superscript in the same column mean no significant difference (*p* > 0.05), but values with different lowercase superscript mean significant difference (*p* < 0.05).

**Table 3 animals-13-02781-t003:** Effects of GSPs on fresh semen quality of bulls.

Grape Seed. Proanthocyanidins	Motility (%)	PR.MOT (%)	Deformity Rate (%)
Before	67.10 ± 2.91 ^b^	56.90 ± 2.29 ^b^	26.07 ± 3.64 ^a^
After	77.00 ± 3.61 ^a^	66.03 ± 2.05 ^a^	18.07 ± 1.82 ^b^

Notes: Values with same lowercase superscript in the same column mean no significant difference (*p* > 0.05), but the values with different lowercase superscript mean significant difference (*p* < 0.05).

## Data Availability

All authors have reviewed and agreed with the published version of the manuscript.

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
