# Peer review of "Grape Seed Proanthocyanidins Improve the Quality of Fresh and Cryopreserved Semen in Bulls"

_animals, 2023, doi:10.3390/ani13172781_

Round 1
Reviewer 1 Report
The work is interestin but is limited in originality and has several methodological issues, but the manuscript could be revised, the content and the aim of the work could be reformulated to focus on “Effects of grape seed proanthocyanidin (GSP) on bull post-thaw sperm quality, when used a feed additive or added to sperm extender”.
Here are some specific comments and suggestions:
1.Simple summary: I suggest revision to replace bold statements which use words “Best” and “discovery”
2. Abstract: Include the duration of the feeding experiment and some data of body weight and feeding rate.
INTRODUCTION
3.The introduction is adequately supporting the aim of this work, it provides a good overview of the potential benefits of GSPs for improving semen quality in bulls but the authors appear to ignore that research on using antioxidants as feed additives or as semen dilutents is extensive, their work should review these gathered knowledge and not present one antioxidant obtained from grape seeds (grape seed proanthocyanidin) as a new tool because it is not.
4.However, there are numerous reports of research work which investigated the antioxidant properties of GSPs on sperm in vivo and in vitro, using it as feed additive or as sperm dilatants. The authors should present the current state of the art and gradually build their argument for the need to do this work and the basis for formulating their hypothesis.
5. The last paragraph of the introduction needs revising, to clearly explain what the pilot study was and what results of the pilot study lead to the formulation of the current hypothesis and aim of the present work.
METHODOLOGY
6. SPERM FREEZINF protocol. The Absract refers to using frozen-thawed sperm stored in liquid nitrogen for 7 days, but the methodology does not describe the sperm freezing and sperm thawing method.
7. The authors used an extender which was based on sodium citrate, TRIS, glucose, glycerol and egg yolk. However, they did not provide details of the extended and the experimental dilutions of the differenct concentration of GSPs tested here (such as pH, osmolarity) a reference to support the choice of this extender should also be included in the methodology. It would be helpful to know if this extender has been shown to be effective in improving sperm quality in bulls, as glycerol may cause osmotic stress and toxicity https://doi.org/10.3389/fvets.2019.00268
8. The authors also hypothesized that GSPs antioxidant properties are significant. However, they did not test their hypothesis by testing another antioxidant. This would have been a good way to confirm their hypothesis on the significant properties of grape seed proanthocyanidins (GSPs) on sperm.
9. FEEDING of the Animals. There is no information on the feed used and the differences between the control and the experimental animals in terms of the feed composition and feeding regime. Some commercial feeds for bulls can contain some antioxidants so data on the antioxidant capacity of the control and the experimental feeds is also crucial.
10. Line 156. The -20 C temperature is not sufficient cold to preserve the activities of the investigated enzymes. This makes this part of the research work not valid, and all these results should be viewed with caution, even better removed from the manuscript.
.
RESULS.
11. Line 175. Revise the heading to explain that you are here presenting the results of using GSPs as antioxidant feed additive.
DISCUSSION
12. The discussion section should start by summarizing the main findings of the study. This would help to orient the reader and provide a framework for the rest of the discussion.
13. The authors should discuss the implications of the findings and the potential mechanisms by which GSPs protect sperm from oxidative damage, as well as the potential applications of other antioxidants for animal reproduction.
- The authors should also discuss the limitations of the study. This would help to put the findings in context and suggest further research.
- The writing should be more concise and to the point. Some of the sentences are long and could be broken up into shorter sentences.
- The writing should be more specific. Some of the statements are general and are not followed by specific examples of the present work.
14. Some of the statements are bold and subjective, and the use of first-person pronouns should be avoided. For example, the sentence "In this study, we found that GSPs had a protective effect in semen motility, viability, mitochondrial activity, plasma membrane integrity, acrosome integrity and antioxidant penitential, which partially consistent with previous research" could be improved by making the following changes:
- Remove the first-person pronoun "we." The sentence could be rewritten as "GSPs were found to have a protective effect in semen motility, viability, mitochondrial activity, plasma membrane integrity, acrosome integrity, and antioxidant status."
- Make the statement more objective. The sentence could be rewritten as "These findings are consistent with previous research."
Likewise, the sentence "In addition, breeding management such as feed additive effect sperm quality of male during spermatozoa of production" could be improved by making the following changes:
- Remove the first-person pronoun "we." The sentence could be rewritten as "Breeding management, such as the use of feed additives, can affect sperm quality in males during semen production."
Make the statement more objective. The sentence could be rewritten as "These findings suggest that the use of feed additives with antioxidant properties, can affect sperm quality in males during semen production."
Some of the statements are bold and subjective, and the use of first-person pronouns should be avoided. For example, I suggest to replace bold statements which use words “Best” and “discovery” in the simple summary.
Likewise, in the Discussion, the sentence "In this study, we found that GSPs had a protective effect in semen motility, viability, mitochondrial activity, plasma membrane integrity, acrosome integrity and antioxidant penitential, which partially consistent with previous research" could be improved by making the following changes:
- Remove the first-person pronoun "we." The sentence could be rewritten as "GSPs were found to have a protective effect in semen motility, viability, mitochondrial activity, plasma membrane integrity, acrosome integrity, and antioxidant status."
- Make the statement more objective. The sentence could be rewritten as "These findings are consistent with previous research."
Likewise, the sentence "In addition, breeding management such as feed additive effect sperm quality of male during spermatozoa of production" could be improved by making the following changes:
- Remove the first-person pronoun "we." The sentence could be rewritten as "Breeding management, such as the use of feed additives, can affect sperm quality in males during semen production."
Reviewer 2 Report
Brief Summary:
The authors evaluated the effects of Grape Seed Proanthocyanidins on the quality of fresh and cryopreserved bovine semen. The topic of study is interesting and current. The manuscript provides new insights into the development of new semen diluent additives and dietary additives. However, it has a few weaknesses and issues that should be addressed.
General Comment:
1. In the current study, 20 mg /kg body weight GSPs was used in a basal diet. What is the basal diet component? What is the rationale for selecting 20 mg/kg GSPs? Are you using the same GSPs in in vitro and in vivo experiments?
2. The reviewer wondering why did not determine the ROS generation levels in sperm after GSPs treatment?
3. The reviewer suggests that perform IVF experiments in the next future.
Specific Comments:
1. Page 3 lines 89-91, you said that “Studies have shown that adding an appropriate concentration of GSPs can significantly improving the semen cryopreservation effect of pigs [17,18]”. Please check Ref 17 again, is it relates to porcine sperm?
2. Page 4 line 111, Please provide more information about GSPs, for example Company and Lot No.
3. Figure 3, why did not stain sperm head DNA using H33342? So it is easy to distinguish between the head and the tail.
Round 2
Reviewer 1 Report
I appreciate your efforts to revise your manuscript. I suggest that you remove the "the most" from the first sentence of the simple summary.